# Floating wind turbine motions signature in the far-wake spectral content - A wind tunnel experiment

Benyamin Schliffke[1, 2], Boris Conan[1], and Sandrine Aubrun[1]

[1]Nantes Université, École Centrale Nantes, CNRS, LHEEA, UMR 6598, F-44000 Nantes, France
[2]Agence de l'Environnement et de la Maîtrise de l'Energie (ADEME), F-49000 Angers, France

**Correspondence:** Sandrine Aubrun (sandrine.aubrun@ec-nantes.fr)

**Abstract.** The growing interest in floating offshore wind turbines (FOWTs) is rooted in the potential source for increased offshore energy production. As the technology is still in a pre-industrial state, several questions remain to be addressed where little field data is available. This study uses physical modelling at a reduced scale to investigate the signature of the floating motions into the wake spectral content of a simplified FOWT model. A wind turbine model based on the actuator disc concept
is placed in an atmospheric boundary layer wind tunnel and subjected to a range of surge, heave and pitch motions. The signatures of idealised sinusoidal motion and realistic broad-band motion on the model's wake at distance of $4.6\,D$ ($D$ being the disc diameter) and $8\,D$ are measured through the use of a rake of single hot-wires. The spectral analysis shows that harmonic motion leaves to clear signatures in the far-wake's energy spectra, mainly in the top tip region, while broad-band motion does not leave easily detectable signatures.

## 1 Introduction

Floating offshore wind turbines (FOWTs) have been the focus of extensive research in the recent years, despite being a comparatively young technology. The consequence of the floating motions on the wind turbine steady and unsteady wake behaviour is of primary interest, since the wake interactions between wind turbines organised in a wind farm induce power production loss due to velocity deficit within the wakes and additional fatigue due to wake-added turbulence (Porté-Agel et al., 2019). Up
to now, recent literature has focused on idealised, sinusoidal motion as an approximation for the real motion of an FOWT's platform (e.g. Schliffke et al., 2020; Fontanella et al., 2021; Garcia et al., 2022; Raibaudo et al., 2022). In these experimental studies the authors identify the motion frequencies applied to their respective model FOWTs in the wake of the model turbine. The type of model used (porous disc in Schliffke et al. (2020); Belvasi et al. (2022) and rotating model in Fontanella et al. (2021)) does not affect the identification of the excited frequencies in the wake.

Feist et al. (2021) study the effects of heave and pitch motion on the wake of a model FOWT. They identify the relevant time scale in the wake of their FOWT model to be the period of incoming waves, that drive the pitching motions. The authors state that both constructive and destructive interference is possible between the dominant scales in the wake and a potential downstream turbine, highlighting the necessity to take this potentially dangerous interaction into account when designing

FOWTs. Regarding time averaged mean and turbulent quantities, Feist et al. (2021) and Meng et al. (2022) find no significant
differences between a fixed turbine and an FOWT.

In their study, Fontanella et al. (2022) identify surge motion to produce velocity variations in the near wake at the imposed motion frequency, similar to what Feist et al. (2021) observed for pitch motion. The velocity variations are caused by the variation in thrust force induced by surge motion. The velocity variations identified by Fontanella et al. (2022) are convected downstream by the mean wake velocity.

Li et al. (2022) recently studied the onset of wake meandering when a FOWT is subjected to sway motion. The authors employ an LES flow model. The authors use a uniform and shear-free inflow, applying harmonic sway motion to the FOWT. A linear stability analysis confirms that the most unstable frequencies in the wake are in the range of $f_{red} = 0.1$ and $f_{red} = 0.6$ ($f_{red}$ based on the disc/rotor diameter and the free stream velocity at hub height), with an optimum of amplification at $0.2 < f_{red} < 0.3$ . Li et al. (2022) state the lateral motion of the FOWT can trigger meandering in the far wake, even at small motion
amplitudes (less than $0.01\,D$). The FOWT wake meandering is therefore not only driven by a quasi-static response of the whole wake to the variation of its emission location, but is amplified when the wake's unstable frequencies and the floater's natural frequencies overlap. The authors also find that increasingly turbulent inflow conditions can inhibit far wake meandering. In their LES study of a rotating wind turbine subject to harmonic heave, surge and pitch motion in a uniform inflow, Kleine et al. (2022) found that predictions of linear stability of helical vortices are well retrieved. The highest interaction is found when the
frequency motion is 1.5 times the rotation of the turbine.

From the literature discussed here, it is evident that the non-stationary behaviour of an FOWT's wake is an important field of research. In order to gain an understanding of the actual behaviour, it is necessary to study the non-stationary behaviour of an FOWT subjected to realistic, i.e. broad-band, motion. What is mentioned as a limiting factor in all the studies discussed above is both the use of idealised, sinusoidal motion regimes and the use of uniform inflow conditions. In this study we will
attempt to better understand the differences between imposed idealised and realistic motion on a model wind turbine using a porous disc model in an atmospheric boundary layer wind tunnel. We combine these elements to recreate the most realistic inflow conditions possible.

The present paper is a companion paper of Belvasi et al. (2022) since both works were carried out with the same experimental set-up, except for the instrumentation. Belvasi et al. (2022) deals with "slow" Stereo-PIV acquisition (low temporal resolution
but high spatial resolution) in cross-planes downstream of the model to study the influence of floating motions on the large-scale properties of the wake (instantaneous wake center locations and normalised available power for a virtual downstream wind turbine). Four cases including pitch and surge were tested and they conclude that the frequency characteristics of the harmonic imposed motions are detectable in these physical values, even if the overall wake statistics (velocity deficit, Turbulent Kinetic Energy) are barely modified by the motions. However, the number of cases was limited. Further, the nature of the
instrumentation did not allow for a detailed spectral analysis.

The present study uses a rake of hot-wires (high temporal resolution but low spatial resolution) and focuses on the local signature of the floating motions within the spectral content of the wake velocity fluctuations. The study includes both harmonic heave, surge and pitch motion in a large variety of amplitudes and frequencies together with realistic motions corresponding to

a floating barge for each Degree of Freedom individually and combined. A total of 19 cases are analysed that are considered to
be most representative of aligned wind/wave conditions. The locations in the wake section where the signature of a variety of
motions (Degrees of freedom (Dof), amplitude, frequency) are the most visible are identified. The intensity of the signature is
also quantified for both harmonic and idealised cases.

In Belvasi et al. (2022) as well as in the present study, the modelling concept for the wind turbine is based on the actuator
disc theory, i.e. a porous disc. The porous disc has proven to be a cost-effective and reliable approach to experimentally
generate a wake similar to a wind turbine wake. Aubrun et al. (2013) find that the porous disc model produces a far wake
nearly indistinguishable from the wake of a rotating turbine at $x/D > 3$ downstream from both models, if immersed in a
typical atmospheric boundary layer (ABL). All significant streamwise quantities, the statistical quantities and the integral
length scale are similar beyond this distance and the rotational momentum is not measurable. Several studies showed that tip
vortex signatures at the edges of the wake are undetectable at $x/D > 3$ (Aubrun et al., 2013; Zhang et al., 2012, 2013). The
comparability has been extended to higher order statistics (Neunaber et al., 2021). Camp and Cal (2016) identify a limit of
$x/D > 3.5$ for differences between the wakes of a rotating model and a porous disc to be negligible.

The experimental set-up used in the present paper will be presented in Sect.2.1 where a description will be given on the
wind tunnel scaling law constraints, on the design requirements for the displacement system used to emulate the floating
motions and, on the quality of the modelled atmospheric boundary layer. The measurement protocol, using a rake of single hot-
wires, is described. The matrix of experiments, based on a parametric study of floating motions (Dof, amplitude, frequency,
harmonic/realistic motions), is also presented. In Sect. 3, the signature of imposed motion in 1 Dof only (1-Dof motion;
surge, pitch, heave) is quantified through the identification of added-energy, compared to the fixed configuration, in the energy
spectra of the far-wake velocity fluctuations. Then, the signature of motions imposed simultaneously in 3 Dofs (3-Dof motion)
is investigated. Finally, a conclusion is given in Sect.4.

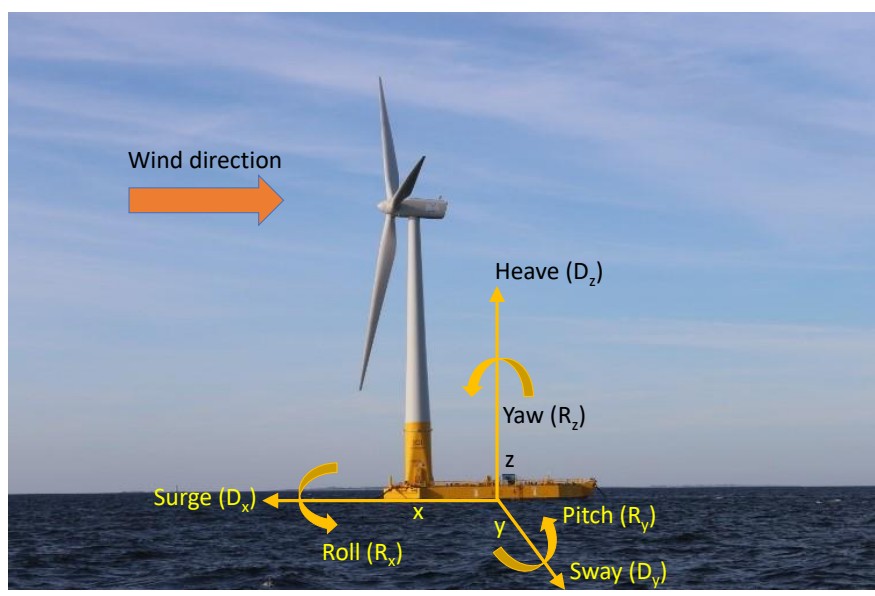

**Figure 1.** Photo of the floating wind turbine FLOATGEN, currently under operation on the sea test site SEM-REV, France. Sketch of Degrees of Freedom (Dof) as used in this work. Longitudinal (x-axis), lateral (y-axis) and vertical (z-axis) translations are referred to as surge ($D_x$), sway ($D_y$) and heave ($D_z$). The rotations around the longitudinal, lateral and vertical axis are referred to as roll ($R_x$), pitch ($R_y$) and yaw ($R_z$). The centre of rotation in the centre of the floater. Adapted from https://sem-rev.ec-nantes.fr/medias/photo/sem-rev-bj-132-bd_1539862739781-jpg?ID_FICHE=196422 Accessed: 2023-11-15

## 2 Experimental Set Up

### 2.1 The reference floating wind turbine

The floating wind turbine presently used as a reference is FLOATGEN (Fig. 1). It is a 2 MW Vestas V80 wind turbine of 80m rotor diameter and 60m hub height installed on a barge, a prototype designed by BW-IDEOL and installed at the SEM-REV test site (Ecole Centrale de Nantes) that has been operational since September 2018. Fitted with a Damping Pool® system, the float measures 36 meters per side and 9.5 meters in height, with a 7.5 meter draught (see BW Ideol, Centrale Nantes (2014) for more information).

### 2.2 Scaling laws in the wind tunnel

The data used in this study is obtained experimentally, working at reduced scale. The scaling laws applied to model the atmospheric boundary layer in the wind tunnel, as well as the reasoning behind the motion regimes employed and the data acquisition are discussed in this section.

In order to maintain comparability between the different motion regimes tested, a reduced frequency ($f_{red}$) is introduced:

$$f_{red} = \frac{f \cdot D}{U_{hub}} \tag{1}$$

where $f$ is the imposed motion frequency, $U_{hub}$ is the wind velocity at hub height $h$ (considered as the physically representative reference velocity for the wake flow scales) and $D$ is the turbine's diameter. From this definition, a higher $f_{red}$ can represent an increased motion frequency, as investigated in the scope of this study, or an increased wind turbine rotor diameter $D$. This cross-dependency justifies the use of higher-than-characteristic $f_{red}$ in the context of future FOWT developments, where much larger turbines, compared to the floater dimensions, are planned (Kikuchi and Ishihara, 2019).

The experiments are carried out at the atmospheric boundary layer wind tunnel of the LHEEA research lab (Ecole Centrale de Nantes, France). The wind tunnel is $24\,m$ long with a $2\,m \times 2\,m$ cross-section. The front section of the wind tunnel consists of the intake, a settling chamber, a honey-comb web followed by a convergent. After a fetch of $19\,m$ the test section begins. A bucket rotor located at the end of the wind tunnel provides the necessary suction. Free stream velocities of up to $10\,ms^{-1}$ are possible.

Scaling factors are influenced by a number of constraints, the size and power of the wind tunnel and Strouhal similarity being the principle factors. According to VDI Guideline 3783 (VDI, 2000), the model must not block more than $5\%$ of the wind tunnel's cross section. Given the wind tunnel's cross section of $4\,m^2$, this results in a maximum disc diameter of $500\,mm$. As the modelled turbine must not be larger than half the expected maximum boundary layer height of around $600\,mm$ to avoid interaction between the wake and the free stream (Schliffke et al., 2020), the top tip of the model must not exceed a height of $300\,mm$. A geometric scaling factor of $\Lambda_L = 1 : 500$ is thus selected, resulting in a porous disc diameter $D$ of $160\,mm$ and a hub height $h$ of $120\,mm$. A velocity scaling factor $\Lambda_v = 3.2$ is introduced to reduce the necessary motion frequencies to a reproducible level.

In order to maintain aerodynamic similarity between flows, Strouhal similarity theory dictates that:

$$S_t = \frac{L_m}{t_m \times U_m} = \frac{L_{fs}}{t_{fs} \times U_{fs}} \tag{2}$$

where $L$, $t$ and $U$ denote a characteristic length, time scale and velocity, respectively, for the model ($m$) and full-scale ($fs$) cases. Employing the scaling factors derived above, a time scaling factor can be calculated:

$$\Lambda_t = \frac{\Lambda_L}{\Lambda_v} \approx 150 \tag{3}$$

As the time scaling factor of 150 indicates, the processes in the modelled flow are 150 times faster compared to the full scale flow. The motion system must thus be capable of reproducing high frequency motion over small distances in a reliable manner. The time scaling factor also implies faster statistical convergence of the average values.

The solidity of the porous media chosen to manufacture the disc is $57\%$. By integrating the velocity profile downstream of the porous disc and applying the momentum theory (Aubrun et al., 2019), the equivalent thrust coefficient can be roughly assessed to $C_T = 0.5$ (Schliffke et al., 2020).

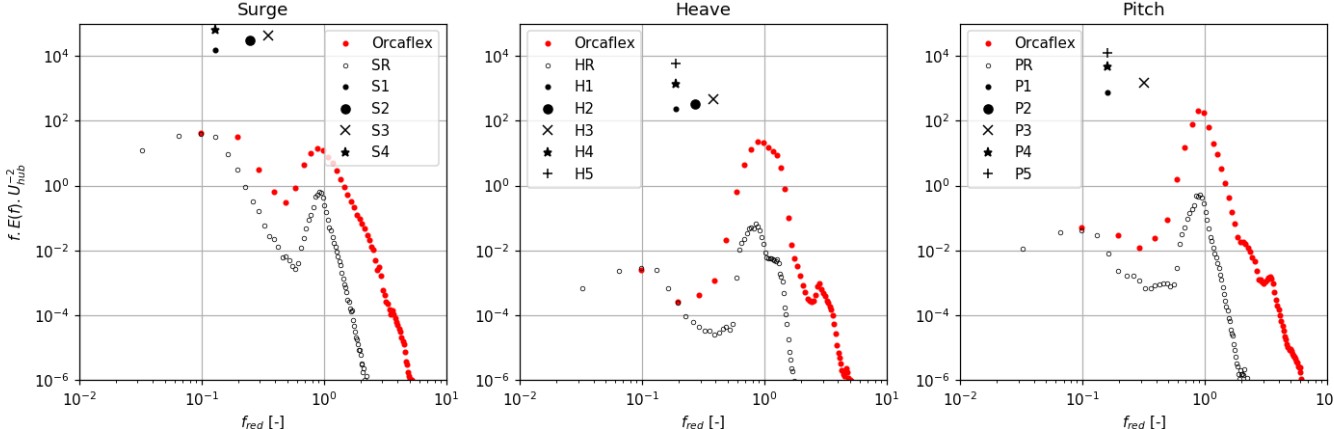

**Figure 2.** Power spectral density of the motions of the FLOATGEN floater ($H_s = 4.25\,m$, $T_p = 11.37\,s$ and wave propagation direction aligned with the wind direction) obtained by Orcaflex simulations and of the motions imposed by the motion system in the wind tunnel. Energy contained in the imposed harmonic motions is added for comparison. a) Surge, b) Heave, c) Pitch.

## 2.3 Motion Modelling

To design the motion system, typical floater motion frequencies and amplitudes are derived from times series coming from a numerical model provided by BW Ideol using OrcaFlex. The time-domain simulations have been satisfyingly compared
to lab and full scale experiments (Choisnet et al., 2018; BW-Ideol, 2019a, b). The sea conditions provided as input to the model are representative of a rough sea state at the test site and correspond to the upper operational boundary of the FOWT. It has a significant wave height of $H_s = 4.25\,m$ with a wave period of $T_p = 11.37\,s$ and the wave propagation direction is aligned with the wind direction. The velocity at hub height that is used to compute the above reduced frequency is chosen as $U_{hub} = 8\,ms^{-1}$. It corresponds to the order of magnitude of the mean wind speed, at $100\,m$, at the sea test site SEM-
REV (Thilleul and Perignon, 2022). These conditions are selected for dimensioning the motion system and as reference for the realistic motion as they induce the largest motions at frequencies that can be safely modelled at reduced scale and that are expected to have an impact on the overall far-wake dynamics, as wake pulsing or meandering ($f_{red} < 0.5$). In the power spectral density of the surge, heave and pitch motion represented in Fig. 2 (OrcaFlex results), two peaks can be identified. The first peak near $f_{red} = 1$ relates to the floater motion induced by wave-to-wave frequencies, it is called here first-order motion.
The second peak at lower frequency, near $f_{red} = 0.1$ is related to the response of the floater linked to the mooring lines and anchoring characteristics, it is called second-order motion. The characteristic second order motions amplitude and frequency for surge are found to be in the range of $A_{fs} = 10\,m$ and $f_{fs} = 0.01\,Hz$ ($fs$ for full-scale). It corresponds to reduced amplitude and frequency in the range of $A_{red} = \frac{A}{D} = 0.125$ and $f_{red} = 0.1$, respectively. The characteristic first order surge amplitudes are at least one decade smaller and their frequency one decade higher, close to $f_{red} = 1$. The latter are therefore not expected
to play a significant role on the global wake dynamics and therefore, will not be taken into account in the motion system design

requirements. Regarding second-order heave and pitch motions amplitudes, they are visible in the spectra (Fig. 2, middle and right, Orcaflex results) but remain small compared to first-order ones. Other Dof motions (sway, roll and yaw) are considered as negligible for the present sea state conditions (wind and wave aligned).

The prominence of three Dofs, i.e. surge, heave and pitch, leads to the design of a 3Dof-motion system. Applying the scaling factors ($\Lambda_L = 500$, $\Lambda_v = 3.2$ and $\Lambda_t = 150$) to the motion delivers amplitudes of several $cm$, whereas the characteristic motion frequencies at model scale lie in the range of $2\,Hz$.

The motion system is composed of three linear motors (including one rotative linear motor), synchronised to reproduce combinations of displacements and rotation in the $(x,z)$ plane. The motion system can reproduce the demanded motions with a temporal resolution of $1\,ms$ and an average precision of $1\%$ for the longitudinal and rotational motions with regard to the commanded position. The average precision is $2\%$ for the vertical axis. A delay of $4\,ms$ is found between the commanded position and the real position of the motion system. The system is controlled using Motion Perfect v5.0.3 software. The system is installed below the wind tunnel floor in a sealed wooden box. No impact of the opening in the floor on the flow within the test section is detected. No propagation of vibrations from the motion system to the measurement equipment is detected.

## 2.4   Data Processing

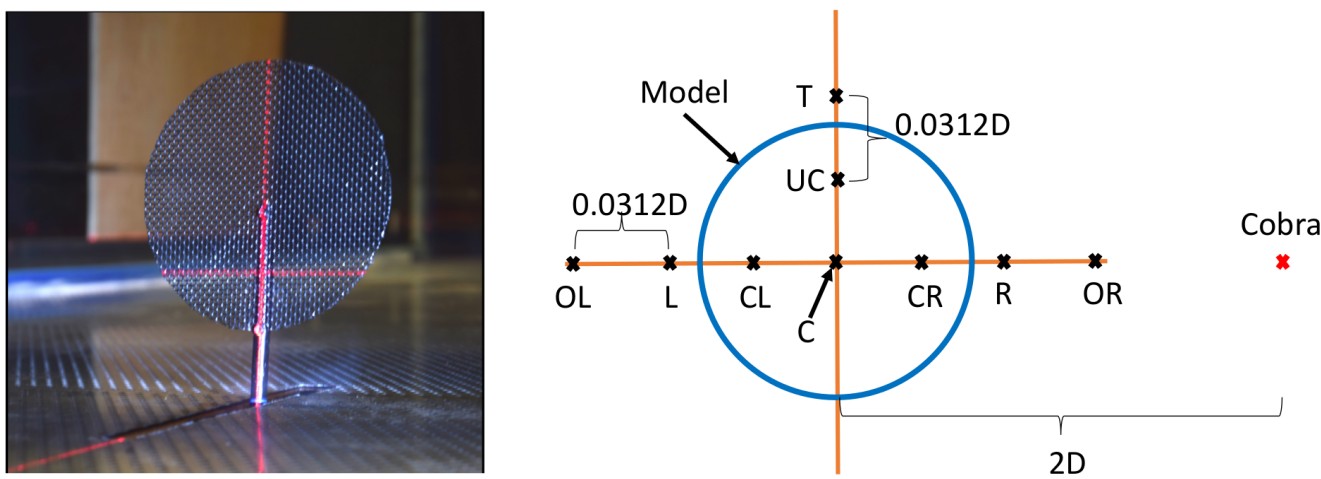

**Figure 3.** Set-up of the wind tunnel experiments showing the porous disc in the test-section. On the right is a sketch of the set up used during hot-wire measurements. $D$ is the diameter of the wind turbine model ($160mm$). The letters are the location codes used in this paper.

Nine single hot-wire anemometers (Dantec Type 55-P11 probe) are arranged in a cross on a rake, as shown in Fig. 3. The rake is placed $4.6\,D$ and $8\,D$ downstream of the actuator disc model. The hot-wires are calibrated in-situ every day. A pitot tube located near the rake during calibration measures the reference velocity. The coefficients of King's Law (King, 1914) are determined using a best-fit approach. Pressure and temperature corrections are applied to the data. The $\chi^2$ goodness of fit value is in the range from $4 \cdot 10^{-4}$ to $1 \cdot 10^{-6}$ for all probes.

The acquisition frequency is at $15\,kHz$ for an acquisition time of $1800\,s$. Outliers ($> 5\sigma$, $\sigma$ being the standard deviation of the acquired time series) are removed from the hot-wire data. In order to limit noise, the signal is filtered at $1\,kHz$ using a digital low-pass Butterworth filter. According to Benedict and Gould (1996), the convergence uncertainty of the mean velocity ($\epsilon_{\overline{U}}$) and standard deviation of the velocity ($\epsilon_{\sigma_U}$) is assessed through 4 and 5, respectively:

$$\epsilon_{\overline{U}} = \frac{Z \cdot I_u}{\sqrt{N_b}} \tag{4}$$

$$\epsilon_{\sigma_U} = \frac{Z}{\sqrt{2N_b}} \tag{5}$$

Where $I_u = \sigma_u/\bar{U}$ is the streamwise turbulent intensity, defined as the ratio between the standard deviation and the time average of a local streamwise velocity and, which for the current study is assessed to $25\%$ maximum. $Z$ is the parameter of the confidence interval (1.96 for the confidence interval of $95\%$), $N_b$ is the number of independent samples, assessed by the ratio between time series duration of 1800s and twice the integral time scale (chosen here as the largest time scale of all motions, 0.5s). The maximum relative uncertainty in the mean velocity and its standard deviation is around $1\%$ and $3.3\%$, respectively.

The mean power spectral density (PSD) of the velocity measurements is calculated using Welch method. The applied window size is set to approximately $4.4\,s$ with a $50\%$ overlap, leading to a mean PSD $E$ built with the average of approximately 800 spectra of $0.3\,Hz$ frequency resolution. The normalised pre-multiplied PSD is then defined as $\frac{f \cdot E}{\sigma_{U_\infty}^2}$, where $f$ is the frequency, $E$ is the mean PSD and $\sigma_{U_\infty}$ is the standard deviation of the velocity measured in the free stream.

The harmonic motion data is treated using a simple Fast Fourier Transform ($fft$) function, as the Welch's method is not applicable to pure harmonic signals.

When searching for the motion's signature in the normalised pre-multiplied PSD in the wake of an FOWT, a comparison between the fixed turbine and the moving turbine is essential. To quantify these effects we calculate the added normalised energy as the difference between the normalised pre-multiplied energy spectra:

$$\varphi = \frac{f \cdot E_{moving}}{\sigma_U^2} - \frac{f \cdot E_{fixed}}{\sigma_U^2} \tag{6}$$

This method enables us to detect small differences between the spectra. We define $\varphi_{max}$ as the largest difference between two spectra in the frequency range investigated ($f_{red} \leq 0.5$). We consider $\varphi_{max}$ to be significant when it is greater than 0.05. This value is determined empirically as it captures all clear and few spurious peaks.

## 2.5 The Modelled Boundary Layer

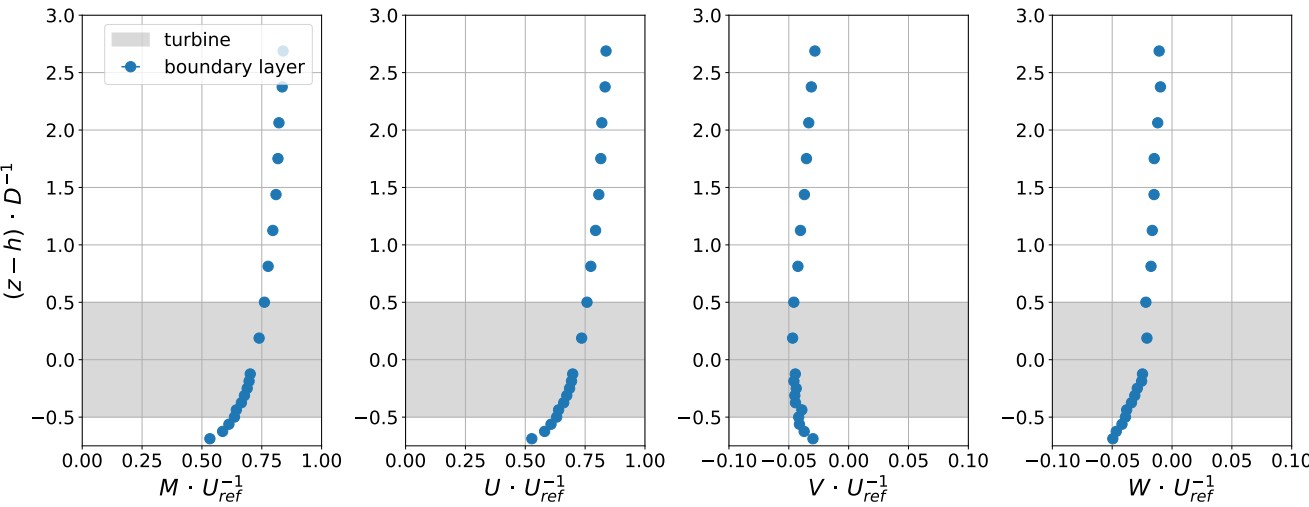

**Figure 4.** Normalized mean velocity profiles for the mean velocity $M$, the longitudinal flow component $U$, the lateral component $V$ and the vertical component $W$ measured in the boundary layer flow. The data is normalized using the averaged free stream velocity $U_{ref}$. Error bars are added, but not visible due to their small values.

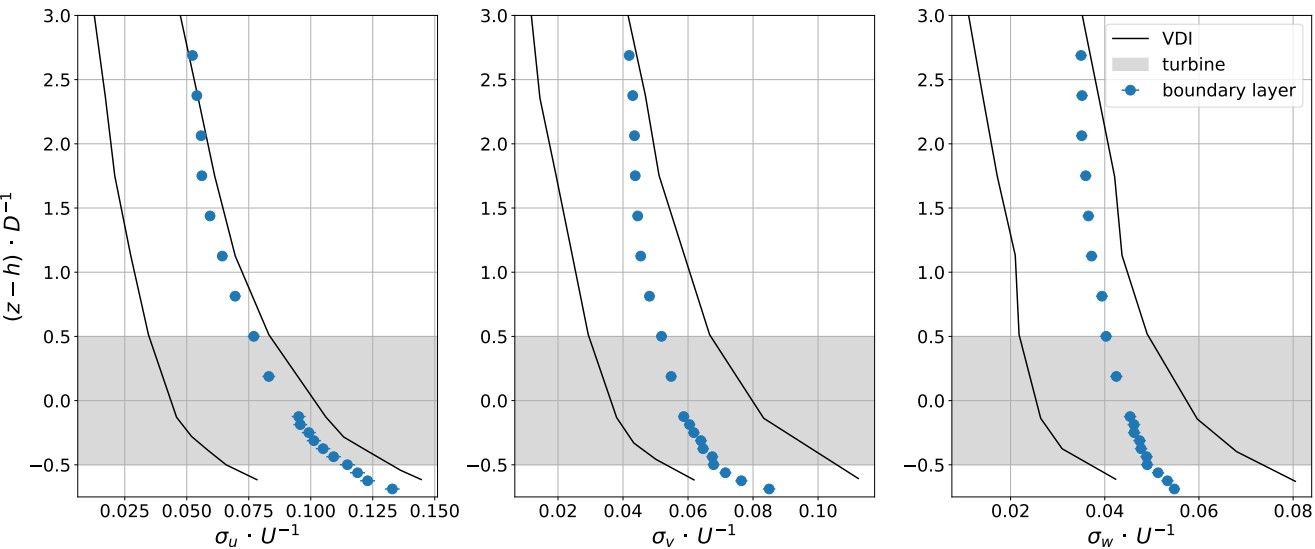

**Figure 5.** Turbulence intensity profiles for the longitudinal flow component $\sigma_u$, the lateral component $\sigma_v$ and the vertical component $\sigma_w$ measured in the boundary layer flow. Error bars are added, but not visible due to their small values.

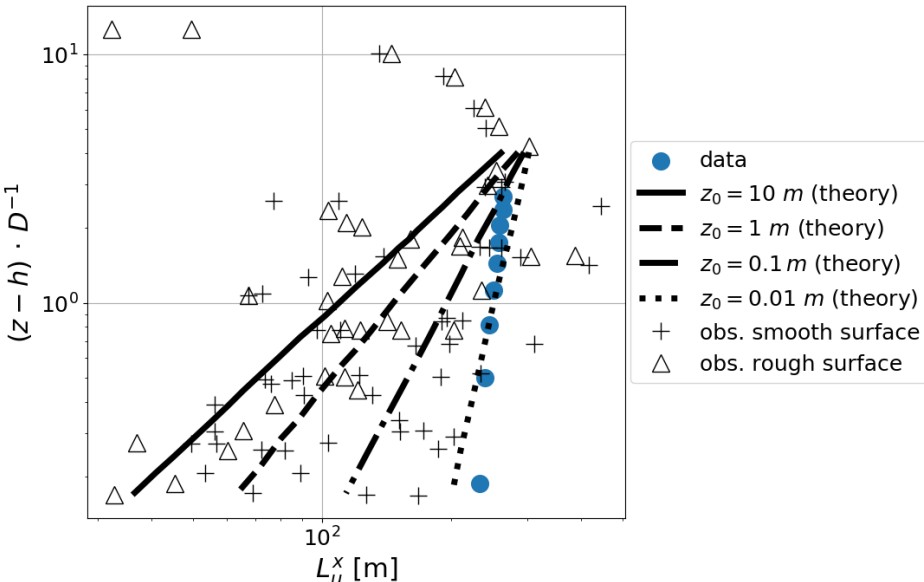

**Figure 6.** Profile of the integral length scale $L_u^x$ in $[m]$ (blue points). Reference data from Counihan (1975) presented for orientation (black lines show theoretical values, crosses and triangles represent field data). Error bars are added, but not visible due to their small values.

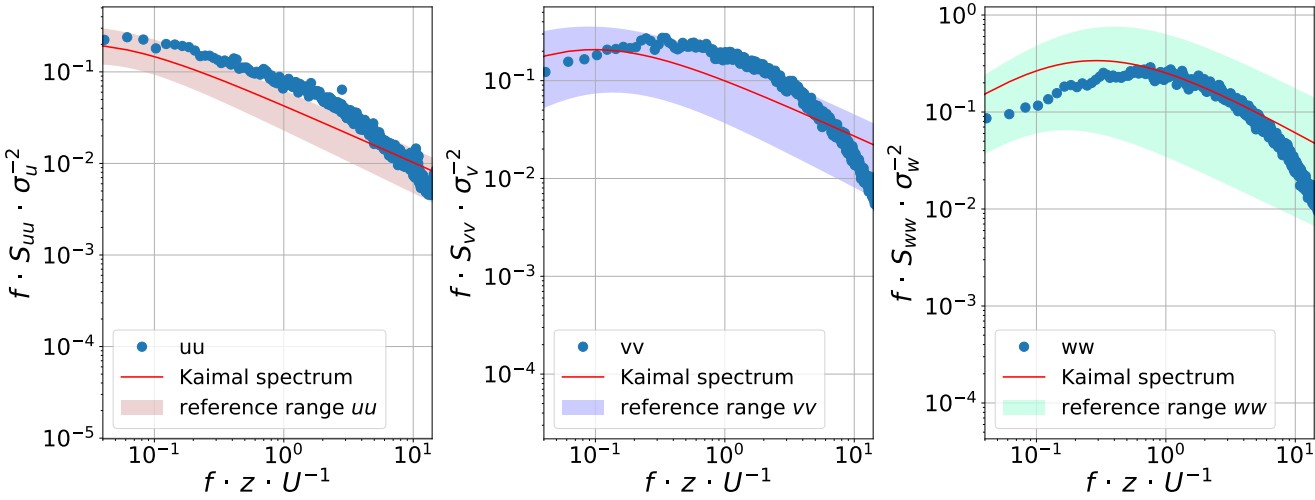

**Figure 7.** Pre-multiplied power spectral density of $U$ (left hand panel), $V$ (central panel) and $W$ (right hand panel). The shaded areas show the range of available reference spectra provided in VDI (2000). All spectra are shown against a reduced frequency.

The modelled atmospheric boundary layer is developed following the VDI Guideline 3783 (VDI, 2000) for a slightly rough surface. The set up is identical to that used in Schliffke et al. (2020). The normalised mean velocity profiles of each flow component are shown in Fig. 4. Notice, that the free-stream velocity ($U_{ref}$) used for the normalization is well above the

wind tunnel boundary layer, near the middle of the test section. For this reason $U/U_{ref}$ does not reach 1. The free stream velocity measured outside of the boundary layer with a Pitot tube was $3\,ms^{-1}$ ($U_{ref}$). The corresponding power-law profile exponent $\alpha$ is 0.11, which is within the proposed range between 0.08 and 0.12 proposed by (VDI, 2000). Using a logarithmic law fitting, the roughness length ($z_0$) at wind-tunnel scale is estimated to be $z_0 = 1.15 \times 10^{-5}\,m$ and the friction velocity to $u^* = 0.12\,ms^{-1}$. At full scale, it gives $z_0 = 5.7 \times 10^{-3}\,m$ that is very close to the VDI Guideline's range [$1 \times 10^{-5}\,m$ - $5 \times 10^{-3}\,m$]. This evaluation differs from Schliffke et al. (2020); Schliffke (2022) were $u^*$ were evaluated separately leading to lower confidence. The lateral flow component $v$ is near zero, as would be expected but consistently negative, indicating a slightly skewed flow to the left, facing downstream. This is probably due to inhomogeneities in the wind tunnel test section. The vertical flow component $w$ is consistently slightly negative, as expected in an ABL. The turbulence intensity profiles for each flow component are in the range suggested by VDI Guideline 3783 (figure 5).

The integral length scale $L_u^x$ profile is shown in figure 6. The measured $L_u^x$ profile corresponds to the theoretical data for very small $z_0$. At $z = 100\,m$ the VDI guideline indicates a range of $L_u^x$ from $200\,m$ to $250\,m$ (full-scale). The wind tunnel data delivers $L_u^x \approx 200\,m$ (full-scale) at the equivalent height, thus matching the VDI's reference range.

To assess the quality of the spectral content of the boundary layer flow, the Kaimal spectrum is used as a reference (Kaimal et al., 1972). We achieved a similar shape and slope in the modelled boundary layer for all flow components (Fig. 7). The measured spectra have a declining slope parallel to the Kaimal spectrum (red line), implying a decay rate of the expected -2/3 slope for pre-multiplied PSD. Nevertheless, even if the experimental spectra are within the range of the reference spectra provided in VDI (2000) (shaded zones), they are shifted towards higher frequencies compared to the Kaimal model. The shift implies that, at wind-tunnel scale, the peak of energy happens for slightly smaller turbulent structures than expected in real life, but integral scales are well scaled and deviations are too limited to expect an impact on the physics described in this paper.

To summarize, the different quantities tested for the modelled atmospheric boundary layer can be considered representative of the natural ABL over a slightly rough surface according to VDI Guideline 3783 (VDI, 2000). The modelled ABL can thus be considered similar to the near-neutral ABL above the sea's surface. Using the motion system described above and the modelled ABL enables us to realistically model the motion of an FOWT.

## 3 Results and discussion

The 19 motion configurations applied for the parametric study are shown in Table 1. It is based on the analysis performed in Sect. 2.3 of the main floater motions (dominant Dofs and realistic motion frequencies and amplitudes). Surge ($R_x$), heave ($R_z$) and pitch ($M_y$) motions are tested, separately or simultaneously ($\sum$). Harmonic motions, with reduced frequencies in the range of realistic frequencies and amplitudes and higher ($0.1 < f_{red} < 0.4$ and $0.0125D < A < 0.25D$ for translation and $2° < \theta < 8°$ for rotation, respectively) are chosen. These configurations are called 'idealised'. The harmonic motions are built with the following equation:

$$D_x, D_z, R_y(t) = A\sin(2\pi f t) = A\sin\left[2\pi\left(\frac{f_{red} * U_{hub}}{D}\right)t\right] \tag{7}$$

**Table 1.** Matrix of experiments. 19 configurations are tested. Cases 1 are the characteristic motion regimes. In cases 2 and 3 the frequencies are varied. In cases 4 and 5 the amplitudes are varied. Cases 'real' use realistic motion profiles. $f_{red}$ is the reduced frequency and $A$ is the amplitude of the harmonic motions imposed to the system. The $\sum$ indicates that the 3 Dof motion is comprised to the three preceding entries in the table.

| Case index | Surge ($D_x$) | | | Heave ($D_z$) | | | Pitch ($R_y$) | | | 3 Dof |
|---|---|---|---|---|---|---|---|---|---|---|
| | Case | $f_{red}$ | $A$ | Case | $f_{red}$ | $A$ | Case | $f_{red}$ | $A$ | |
| 1 | S1 | 0.13 | $0.125\,D$ | H1 | 0.19 | $0.0125\,D$ | P1 | 0.16 | $2°$ | $\sum_1$ |
| 2 | S2 | 0.25 | $0.125\,D$ | H2 | 0.27 | $0.0125\,D$ | P2 | 0.26 | $2°$ | - |
| 3 | S3 | 0.35 | $0.125\,D$ | H3 | 0.38 | $0.0125\,D$ | P3 | 0.32 | $2°$ | - |
| 4 | S4 | 0.13 | $0.25\,D$ | H4 | 0.19 | $0.03125\,D$ | P4 | 0.16 | $5°$ | - |
| 5 | - | | | H5 | 0.19 | $0.0625\,D$ | P5 | 0.16 | $8°$ | - |
| 'real' | SR | | | HR | | | PR | | | $\sum_R$ |

with a mean free stream velocity at hub height $U_{hub}$ of $2.5\,ms^{-1}$, on average.

The choice of the frequency range is driven by (i) the literature review that showed a wake response to motion excitation for reduced frequencies higher than $0.1$ and (ii) the motion system frequency capacity. It leads to a frequency range of $2-6\,Hz$.

The original time series of floater motions are also applied to the system, after a down-scaling to the wind tunnel scale and a low-pass filtering for reduced frequencies higher than $0.1$. This more realistic configuration is called 'real' in the Table 1. The filtering was applied to focus on the low frequency motions due to second order floating effects and to respect the limitations of the motion system. Figure 2 shows the comparison between the power spectral density of the original full-scale surge, heave and pitch motions obtained by time domain simulations with Orcaflex (see 2.3) with the ones of the imposed motions in the wind tunnel experiments for these 'real' configurations. The spectral energy distributions of the realistic surge and pitch motions are well reproduced with the motion system, at the cut-off frequency of the low pass filter. The latter figure shows also the energy contained in the imposed harmonic motions. It is clear from this comparison that the idealised 1-Dof motion regimes (cases 1 to 5) supply to the system much more energy density in the specific frequency range than the realistic regime ( case 'real').

## 3.1 1-Dof idealised and realistic motions

In the following, the normalised energy spectra of the velocity fluctuations measured by the single hot-wires in the wake area is analysed. The spatial distribution and strength of the energy spectra's peak is discussed. First the results for idealised motion regimes are presented, before moving on to realistic 1-Dof motion. Both data measured at $4.6\,D$ and $8\,D$ downstream are discussed. Figure 8 shows an example of the pre-multiplied power spectral density of velocity fluctuations at a downstream distance $4.6\,D$ and different crosswise locations for the fixed configuration and the surge case S3. The overall spectral content is barely modified by the motion effects. This confirms previous results showing that the turbulence intensity, or the Turbulent Kinetic Energy, is slightly affected by the imposed motions (Schliffke et al., 2020; Schliffke, 2022; Belvasi et al., 2022). A

distinct peak is visible at the motion frequency at all locations though, illustrating a transfer of energy from the disc motion to the wake flow, which remains linear up to the far-wake. As described in Sect. 2.4, a systematic analysis of this signature is performed through the detection of the maximum added normalised energy between fixed configuration and moving ones : $\varphi_{max}$.

Figure 9 shows the spatial distribution of $\varphi_{max}$ for surge case S3. Each circle represents the respective measurement location (Fig. 3). The colour indicates the strength of $\varphi_{max}$ at the given location. The points along the vertical center line (C, UC and T) are most clearly affected by the imposed surge motion, with the highest point (T) having the highest $\varphi_{max}$. Further, the outer points at OL and OR have slightly increased $\varphi_{max}$ compared to their respective neighbours. At these positions, the peak emerges more distinguishably from the local turbulence. This proves that these positions are located at the outer edge of the

wake, where the turbulence variance decreases to tend to the external ambient turbulence, whereas the motion imposed to the overall wake is still energetic. Similar behaviour can be observed for the other motion regimes tested in this study, thus we will focus on the vertical centre line for the following analysis.

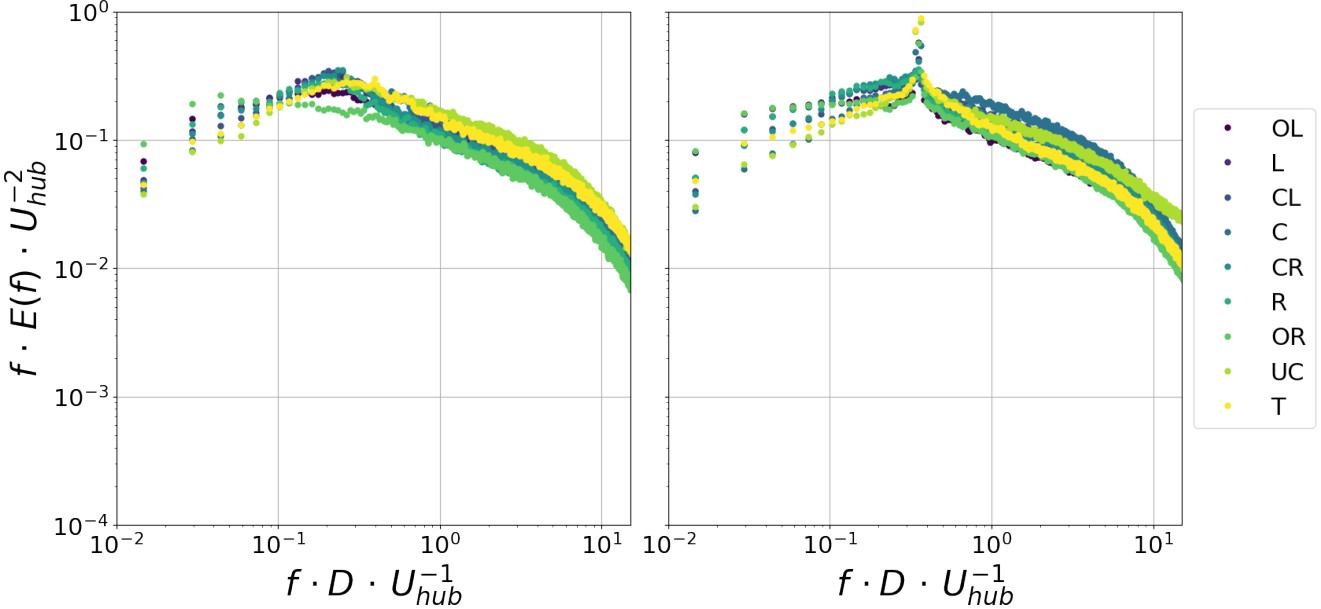

**Figure 8.** Pre-multiplied power spectral density of velocity fluctuations at a downstream distance $4.6\,D$ and different crosswise locations (see Fig. 3) for the fixed (left) and for the imposed idealised surge motion S3 with $f_{red} = 0.35$ with $A = 0.125\,D$ (right) configurations.

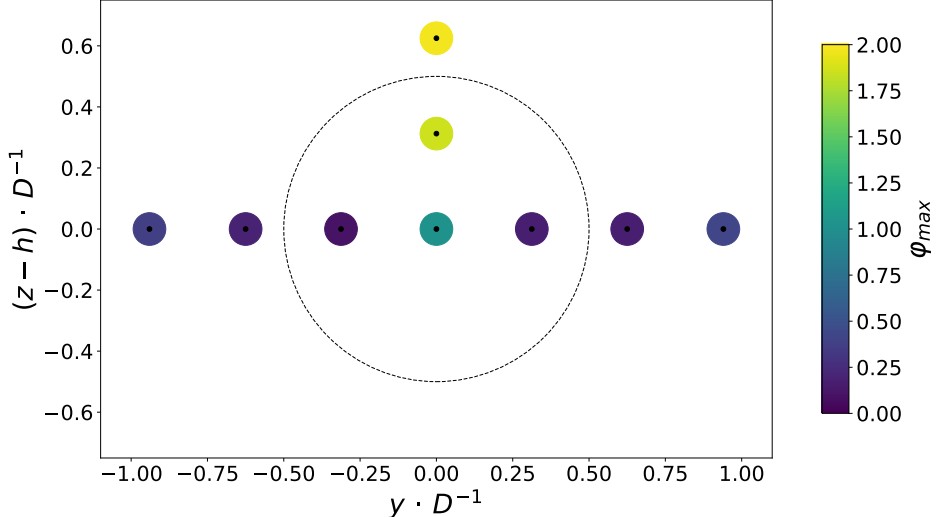

**Figure 9.** $\varphi_{max}$ for imposed idealised surge motion S3 with $f_{red} = 0.35$ with $A = 0.125\,D$ at $4.6\,D$. The colours represent the strength of $\varphi_{max}$.

As Fig. 10 shows, imposed idealised 1-Dof motion at a constant amplitude and varying motion frequencies (cases 1, 2 and 3) does not provide distinguishable peaks for all regimes. For instance, the idealised cases (S1, H1 and P1) do not leave any signature in the wake spectral content. For S1, it is to be noted that there is a peak, but it is not shown as it falls below the threshold imposed. For configurations with a visible peak, it is at the assigned motion frequency, indicating that the energy transfer from the disc motion to the wake dynamics follows a linear process for all these configurations. Figure 10 shows also that the wake dynamics is more receptive to the surge motion and the higher the excitation frequency is, the higher the peak of added normalised energy is. In the limited number of configurations tested, we see a higher signature in the highest range tested close to $f_{red} = 0.3$. Very few results are available in the literature to compare but, in their study of sway motion with idealized inflow conditions, (Li et al., 2022) showed a maximum of receptivity also in the range [0.2-0.3], but the local maximum they found is not visible in the present case.

No peak is visible for heave and pitch motions at low frequencies ( H1, H2, P1 and P2) but one slightly emerges at the higher frequency for both motions (H3 at $f_{red} = 0.38$ and P3 at $f_{red} = 0.32$). The heave and pitch motions have ten times smaller amplitudes compared to surge one. This can explain the weak spectral signature of these motions in the wake dynamics. The detected peaks for surge motion (left hand panel) also show a dependency in height, as the clearest peaks are detected around the top tip of the turbine model. Similar observations can be made for heave (central panel) and pitch motion (right hand panel). At $8\,D$ the peak magnitudes are roughly halved for surge motion. The peak for S1 is now visible, this is in agreement with (Belvasi et al., 2022) where a peak is visible in the PSD of porous disc wake power for case SI where (0.125D/0.11), which is very similar to our case S1. At that distance, no peaks are detected for heave and pitch motions anymore (Fig. 11). In the present study, the pre-multiplied PSD of the longitudinal wind speed component is analysed. In contrast, in (Belvasi et al.,

2022) the PSD of the porous disc wake power is shown, which comprises spatial information of the wake and is linked to the velocity at power 3. Therefore, a direct comparison with the present work is not straightforward.

When varying the amplitude of the motion at a constant reduced motion frequency (Fig. 12) at $4.6\,D$, the tested motion regimes with increased amplitudes deliver increased peaks at the assigned frequency (cases 1 and 4). The wake response to the motions remains linear. For surge motion (left hand panel) the lowest peak is located above top tip, whereas it is the location of the highest peak for both heave regimes. Considering the scatter that one can observe on a power spectral density in turbulent flows and so, the variability of the indicator $\varphi_{max}$, no attempt of interpretation of this opposite trend is presently performed.

For pitch motion, the tested motion regimes do not deliver a visible peak in the normalised energy spectrum. In case P5, that is similar to the case PII in (Belvasi et al., 2022) where a peak in the power is detected, the peak is too weak to be detected. The explanation is that pitch motion representative of the floater induces a translational motion of the wind turbine model, which is comparable to surge motion. The chord of the rotational motion by the top tip is thus considered as equivalent to the amplitude of surge motion. The maximum amplitude of $8°$ thus gives an equivalent surge motion amplitude of approximately $0.063\,D$. This amplitude is about half of the minimal amplitude investigated when analysing surge motion $(0.125\,D)$, that also shows no clear peaks. Thus the limited effects of pitch motion on the wake are logical. At $8\,D$ the peaks are about half as strong as at $4.6\,D$. For heave the motion regime in case 4 does not deliver a detectable peak anymore.

As already mentioned, the idealised characteristic 1-Dof motion regimes (cases 1: S1, H1 and P1) do not deliver any detectable peaks in $\varphi$. The same is true for the realistic 1-Dof motion regimes (case 'real': SR, HR, and PR). The absence of distinct harmonic signature is expected since the realistic motion presents a broadband spectrum, but it does not show any significant modification of the overall spectral content neither. The broadband energy transferred by the disc motion to the wake flow is not strong enough to leave a signature in the far-wake flow. Associated plots are therefore not displayed.

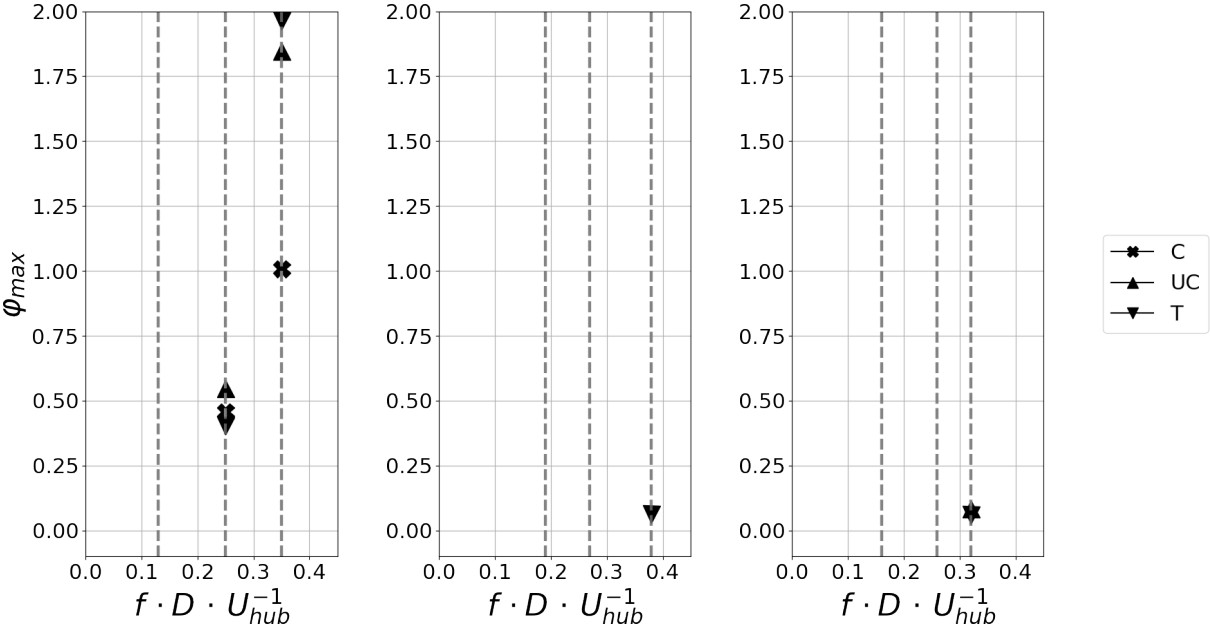

**Figure 10.** $\varphi_{max}$ for imposed idealised motion with a constant amplitude ($A = 0.125\,D$) and different reduced frequencies (cases 1, 2 and 3) measured at $4.6\,D$ downstream of the wind turbine model. Surge S1, S2 and S3 (left hand panel), heave H1, H2 and H3 (centre panel) and pitch P1, P2 and P3 (right hand panel). Only strictly positive $\varphi_{max}$ values are plotted: S1, H1, H2, P1 and P2 configurations are therefore not visible.

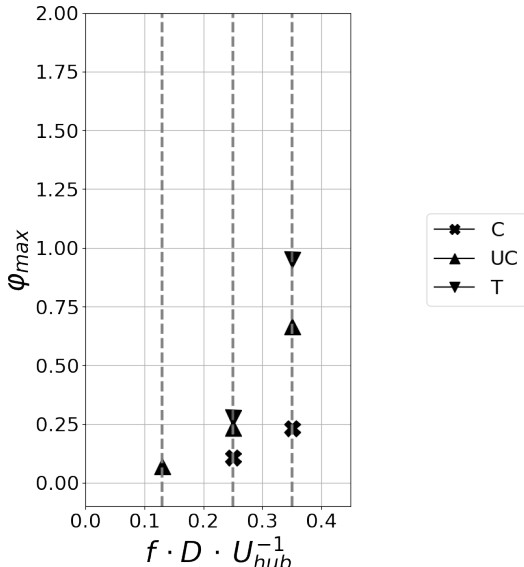

**Figure 11.** $\varphi_{max}$ for imposed idealised surge motion with a constant amplitude ($A = 0.125\,D/2°$) and different reduced frequencies (cases S1, S2 and S3) at $8\,D$ downstream of the wind turbine model. Only strictly positive $\varphi_{max}$ values are plotted.

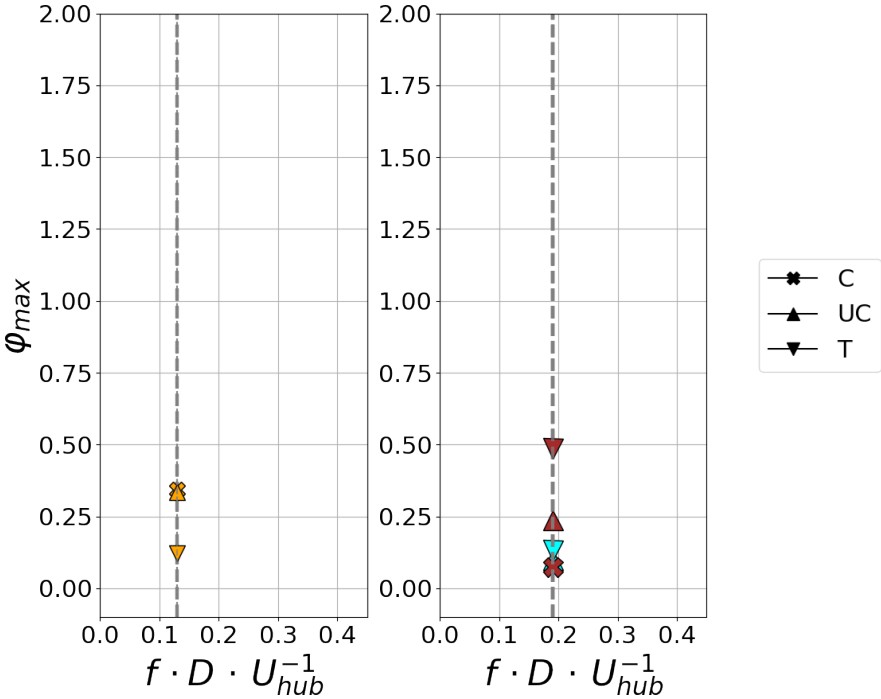

**Figure 12.** $\varphi_{max}$ for imposed idealised motion with different amplitudes (cases 1, 4 and 5) at $4.6\,D$ downstream of the wind turbine model. Surge S1 and S4 (left hand panel, S4: orange) and heave H1, H4 and H5 (right hand panel, H4: blue, H5: brown). Only strictly positive $\varphi_{max}$ values are plotted: S1 and H1 configurations are therefore not visible.

## 3.2  3-Dof idealised and realistic motions

In this section the wake's response to 3-Dof motion is investigated. The idealised 3-Dof motion regime consists of the combination of characteristic harmonic motions for surge, heave and pitch (surge: $f_{red} = 0.13; A = 0.125\,D$, heave: $f_{red} = 0.19; A = 0.0125\,D$, pitch: $f_{red} = 0.16, A = 2°$). The realistic motion is the combination of the respective 1-Dof motion regimes already discussed above. The main question to be answered in this section is whether the superposition of the single Dof motions has more effect on the wake than each Dof by itself. Figure 13 shows the respective graphs of $\varphi$ for idealised motion (left hand panel) and 'real' motion data (right hand panel) measured at $4.6\,D$.

For both the idealised and realistic motion regimes, $\varphi$ oscillates around zero. Regarding the idealised motion (left hand panel), a weak trace of the surge motion can be seen at $f_{red} = 0.13$. Heave and pitch motion do not leave traces in $\varphi$ that can be separated from the overall noise of the signal. Realistic 3-Dof motion does not leave any noticeable trace in $\varphi$ (right hand panel). In addition, in Belvasi et al. (2022), no evident difference was observed on the TKE. This does not appear surprising, when considering the fact that the characteristic and realistic 1-Dof motion regimes also do not leave a clear trace in energy spectra of the wake.

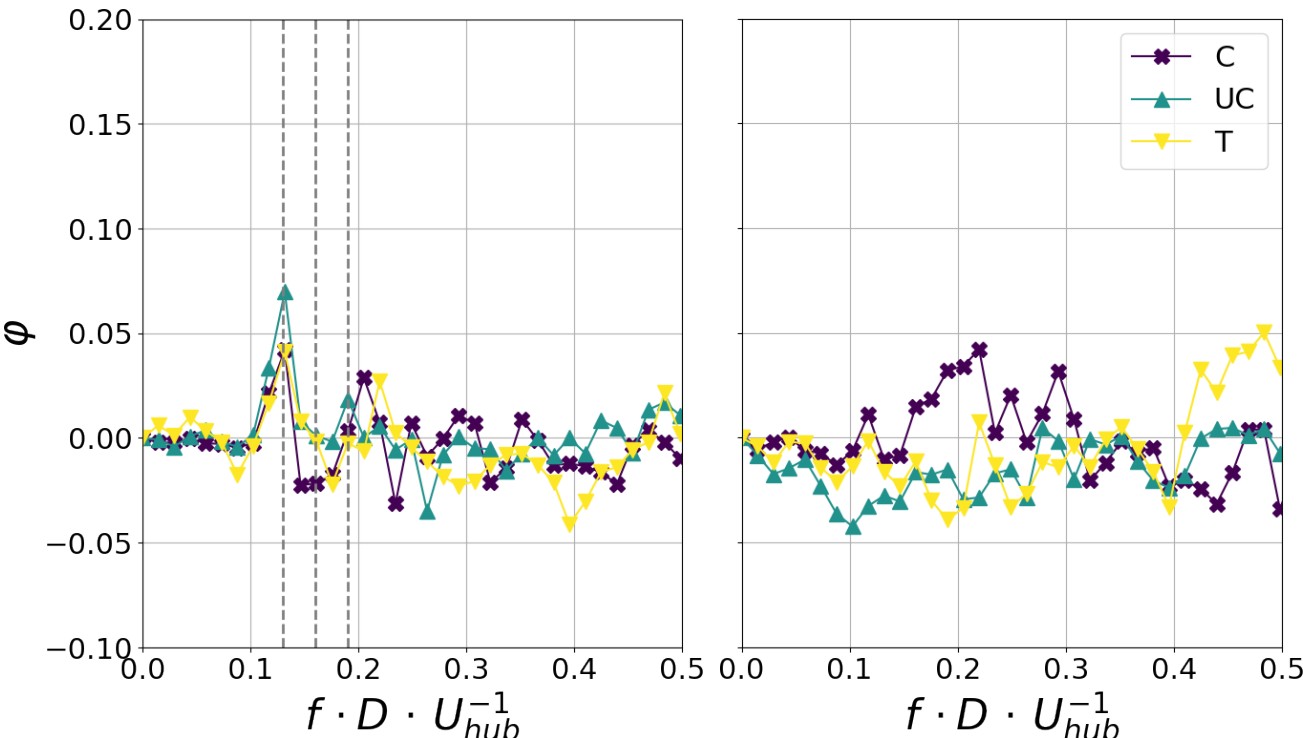

**Figure 13.** $\varphi$ at $4.6\,D$ downstream of the wind turbine model for the idealised 3-Dof motion (left) and the 'real' 3-Dof motion. The dashed lines indicate the imposed frequencies of the 3 respective Dofs.

## 4 Conclusions

In this study a parametric study on the effect of 19 idealised or realistic motion regimes imposed on a model wind turbine immersed in an atmospheric boundary layer were performed experimentally. A porous disc model is subjected to surge, heave and pitch motions derived from and varied around numerically simulated motion data, using the FLOATGEN demonstrator as a full scale reference. A neutral atmospheric boundary layer representative of an offshore environment is modelled in the wind tunnel to the standards suggested by VDI Guideline 3783 (VDI, 2000).

In general, when idealised 1 Dof motions characteristic of the actual floater motions are imposed, they do not leave signature in the energy spectra of a modelled FOWT's wake at $4.6\,D$ downstream of the wind turbine model. One must have either a higher frequency or amplitude in order to leave measurable traces in the energy spectra. Surge, as the main motion component in this study, leaves detectable traces in the wake turbulent flow until at least $8\,D$ downstream from the turbine for a reduced motion frequency above $f_{red} = 0.19$ at an amplitude of $A = 0.125\,D$. At $f_{red} = 0.13$ the imposed motion frequency can be detected when applying an amplitude of $A = 0.25\,D$.

Heave and pitch motions have significantly smaller amplitudes than surge motion, thus they require high motion frequencies to be detectable even at $4.6\,D$. Regarding heave motion, $f_{red} = 0.38$ at an amplitude of $A = 0.0125\,D$ is required to detect the motion frequency in the wake flow. Pitch motion with an amplitude of $2°$ leaves a detectable trace in wake's energy spectrum at $f_{red} = 0.32$. None of the motion regimes tested for heave and pitch leave a detectable trace at $8\,D$. However, (Belvasi et al., 2022) find peaks in the PSD of the wake power for a case similar to P5 suggesting that if no clear peak is visible in the vertical centerline, the wake is still affected and more analysis using other approaches may be needed such as high-speed PIV, conditional averaging or phase averaging. The realistic motion regimes employed in this study also do not leave detectable traces in the wake's energy spectrum.

In general, if a peak is detected in the added normalised energy, its frequency is always in agreement with the motion excitation frequency, indicating that the energy transfer from the disc motion to the wake dynamics follows a linear process for all the tested configurations.

Additionally, the conclusions provided by Li et al. (2022) for side-to-side motions are presently confirmed for fore-aft (here, streamwise) and vertical motions. In general, the frequency range $0.2 < f_{red} < 0.3$ leads to the most detectable signature in the energy spectra, illustrating an amplified response of the wake when the wake's unstable frequencies and the floater's natural frequencies overlap. However, contrary to Li et al. (2022), no peak in the response was detected and further experiments would be necessary to reveal if the response keeps increasing or decreases. The consequences of this feature on the FOWT wake meandering are partially studied in Belvasi et al. (2022) with the same experimental set-up.

Superposing the 1 Dof motion regimes to generate a 3 Dof motion does not leave a clear trace in the wake for realistic motion. The idealised motion shows a weak trace at $f_{red} = 0.13$, the frequency of the imposed surge motion. The lack of response from the longitudinal flow component of the wake overall, is most likely explained by the lack of energy in the 3 Dof motion regimes at a given frequency. Thus the given frequencies, or frequency ranges in the case of realistic motion, are not sufficiently excited to be noticeable.

Considering the significant differences found between the effects of imposed realistic and idealised motion on the wake's energy spectrum it may not advisable to use harmonic motion as a proxy for realistic motion regimes. While the idealised motion leaves clear traces in the wake at the imposed motion frequency, the broad band, realistic motion does not seem to have much effect at all. This complements the findings of (Belvasi et al., 2022) where realistic motion is shown not to affect dramatically the wake statistics of the wind speed, the TKE, the wake center nor the wake center distribution. Also, for the motion of a turbine to have an identifiable effect on the wake's energy spectrum the motion must be strong. This means that only really rough sea conditions will have a measurable effect on the energy spectrum of said floating turbine's wake. On the other hand, the next-generation of floating wind turbines, will have dimensions much larger than the present one ($D = 240m$ for a $15MW$ wind turbine instead of $D = 80m$ for a $2MW$ one). With similar environmental conditions and floater stability, the associated reduced frequencies will increase and could fall into a frequency range that is more receptive for the unsteady wake response. The present results are therefore not universal and do not conclude a general absence of floating motions signature in the far-wake of the wind turbine, when motions properties are realistic.

The next steps necessary to further improve our understanding of the impacts of imposed turbine motion on its wake's behaviour are to determine the transfer function between the turbine's motions and the wake's energy spectra, as well as the further description, quantification and modelling of the floating-added wake meandering process. This would allow for, at the end, the calculation of FOWT's wake characteristics early in its design phase, allowing for early optimisations, where possible.

FOWT-specific control strategies could also benefit from improved understanding of the relationship between the turbine's motion and its resulting wake.

*Data availability.* Raw data of each figure is available upon request

*Author contributions.* Conceptualization, B.S., B.C. and S.A.; experiment, B.S., B.C.; data curation and treatment, B.S.; data analysis, B.S., B.C. and S.A.; funding acquisition and project administration, S.A.; supervision, B.C. and S.A.; original draft writing, B.S.; writing—review

and editing, S.A, B.C. and B.S. All authors have read and agreed to the published version of the manuscript.

*Competing interests.* One of the co-authors is a member of the editorial board of Wind Energy Science.

*Acknowledgements.* This work was carried out within the framework of the WEAMEC, West Atlantic Marine Energy Community, and with funding from the Pays de la Loire Region and Europe (European Regional Development Fund). The authors would like to thank Agence de l'Environnement et de la Maîtrise de l'Energie (ADEME) for co-financing Benyamin Schliffke's PhD and thus contributing significantly to

this publication. We would further like to thank BW Ideol for their support in providing simulation data on the FLOATGEN floating motions. We acknowledge Thibaud Piquet's impressive support in keeping the wind tunnel running and making the motion system operational.

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
