# Peer review of "Floating wind turbine motions signature in the far-wake spectral content - A wind tunnel experiment"

_Wind Energy Science, 2023_

## Referee Comment (RC1)

**General comments:**

"Floating wind turbine motions signature in the far-wake spectral content – A wind tunnel experiment" addresses the important question of the relationship between floating offshore wind turbine (FOWT) motions and wake meandering. Using an actuator disc model subject to idealized and realistic motions in a wind tunnel, the spectral signature of the wake is probed and related to the frequencies of the imposed motions. The authors find that the signature of strong, idealized motions appear in the wake, while more realistic motions do not have an apparent impact. While these findings present a step towards understanding wake meandering for FOWTs, the authors could have elaborated on their results to show what impacts realistic wave motions do have on the wake, even if the expected spectral signature was not observed. In addition, some of the details of both the methods and results were not clearly presented. Please see below for specific comments and some minor technical corrections.

**Specific comments:**

1. Page 2, lines 44-46: Can you elaborate on how your work compares with Belvasi et al. (2022)? For example, page 16, line 288 states that none of the pitch regimes tested in the current study leave a detectable trace at $8D$, but figure 13 of Belvasi et al. (2022) shows a clear peak for the case that is similar to P5 in the current study.
2. Page 3, lines 61-62: Camp and Cal (2016) do not show wake spectra in their study. Is there evidence showing that the rotational frequency signature does not persist into the far wake?
3. Page 4, lines 100-101: What does the velocity scaling factor of 3.2 correspond to in terms of the full-scale and model velocities used?
4. Page 5, lines 121-129: The use of "first order" and "second order" is not clear in this context.
5. Page 8, figure 3: Why don't the normalized velocity profiles reach 1 at the top of the plot?
6. Page 8, line 175: Does the proposed range of 0.08 to 0.12 come from the VDI Guideline? Please clarify.
7. Page 8, lines 176-177: Why is the flow slightly skewed to the left?
8. Page 8, line 181-182: The theoretical data is for $z_0 = 0.01$ m, which is still multiple orders of magnitude larger than the roughness length calculated in this experiment ($z_0 = 5.5 \times 10^{-6}$ m). How is this comparison justified?
9. Page 10, figure 6: There are no blue dots (shown in the legend) plotted in the rightmost panel of this figure.
10. Pages 10-11, lines 209-210: This sentence is not very convincing as figure 7 shows only one point from Orcaflex below (or at) the cutoff frequency of 0.1 for each plot.
11. Page 12, lines 227-229: Why do the OL and OR points have more energy for the moving turbine than the fixed turbine (i.e., $\varphi_{max} > 0$), and why does this signify the lateral limits of the wake?
12. Page 13, lines 235-237: It does not appear that the results of this study are consistent with those from Li et al. (2021) which show maximum receptivity for reduced frequencies between 0.2 and 0.3. In the current study, the peak frequencies are all above 0.3. Were any higher frequencies tested? Would $\varphi_{max}$ be expected to continue to increase with frequency?
13. Page 15, line 270: Does the realistic motion affect the total energy of the wake?
14. Page 16, figure 12: This figure is very unclear. Are three cases shown on each plot? If so, different symbols/colors should be used to differentiate them.
15. Page 17, lines 305-306: Does the realistic motion have any impact on the wake? Can you compare, for example, a time series of the wake centerline? Or could you refer to previous studies to put this finding in context?

**Technical corrections:**

1. Page 1, line 22: A word is missing here: "…both constructive and destructive *interference* is possible…"
2. Page 5, line 123: If figure 7 is the first figure referenced, maybe it should be figure 1.
3. Page 7, line 168-169: Make sure you are consistent with your use of $\varphi_{max}$ vs. $\phi_{max}$.
4. Figures 9, 10, 11, and 12: The labels on these figures should read $\varphi_{max}$ rather than $\varphi$.

---

## Referee Comment (RC2)

General comment

"Floating wind turbine motions signature in the far-wake spectral content – A wind tunnel experiment" investigates the relationship between a floating offshore wind turbine (FOWT) motion and wake dynamics. The authors consider a realistic FOWT motion, and they want to compare its impact on the wake signature to that obtained using idealized motions (harmonics, etc).

For this research work, they perform wind tunnel experiments with a porous Actuator Disk. The model is equipped with a motor that allows to represent heave, surge and pitch motions of the floater (the other motions are negligible for the considered "wave" case). When idealized motions are considered, the signature of those motions is visible in the spectral content of the wake velocity fluctuations (when high amplitudes and high frequencies are imposed). When a realistic FOWT motion is considered, the impact of this motion on the wake power spectra is negligible. The authors conclude that one should be careful by using idealized sinusoidal signals for the FOWT motions as their impacts on the wake signature are different from those obtained using a more realistic signal (at least for the considered size of wind turbine, i.e. D = 80 m).

The paper is well written and pleasant to read. The research work is very interesting, as it clearly highlights differences in terms of wake response between realistic and idealized conditions. The objectives are clearly highlighted, and the study is very rigorous.

However, they authors could more elaborate on the impact of the realistic motion on the wake dynamics in general. Some paragraphs could also be clearer, and some results should more clearly presented or more discussed. Please see below for more specific comments

Specific comments

1. P2,L30 --- : I am surprise that there are no more LES studies that focus on the impact of the wind turbine motions on the wake signature. Or perhaps, if they do exist, these studies always consider idealized motions?

2. P2, L45 – L50 : comparison with work of Belvasi et al. (2022) The authors have exactly the same set-up? The work of Belvasi et al. (2022) also proposes to investigate differences in wake statistics between idealized and realistic motions? What does this present study add? According to the authors, the work of Belvasi et al. (2022) already highlighted that realistic FOWT motions have an impact on the wake center position, and not on the wake deficit, and on the TKE. The authors should highlight more clearly what this "complementary" research paper will add.

3. P3, L67 -> L70 : In this paragraph, both motions should be mentioned (idealized and realistic). The authors also write that they will only investigate heave, surge and pitch, and it is not clear at this stage why they consider these only.

4. P4-P5, section 2.2 : The authors should give an order of magnitude of the wind speed considered in this research work (both for the wind tunnel and the FOWT).

5. P4-P5 : An image of the experimental set-up (porous disk, …) should be useful (even if it is surely available in other papers).

6. P5, Section motion modeling (2.3) : It is not clear for me : the authors have motion signals coming from Orcaflex, for a particular wave case, and they will use this signal as

a realistic motion ("realistic case" of the paper), right? And, based on this signal, the authors will also isolate characteristic amplitude and frequencies in order to elaborate idealized floater motions? This last part should be more elaborated or described in Section 2.3. This becomes clear later in the paper but this is not the case in this paragraph.

7. P5, L121: What do "first order" and "second order" mean?

8. P8, L175 : "*The corresponding power-law profile exponent α is 0.11, which is within the proposed range between 0.08 and 0.12.*" Where does this "proposed range" come from?

9. P8, L187-188 (discussion of Figure 6) : "*Nevertheless, even if the experimental spectra are within the range of the reference spectra provided in VDI (2000) (shaded zones), they are shifted towards higher frequencies compared to the Kaimal model*". What do the authors conclude about this shift? As the sentence is formulated, the reader expects a conclusion. This shift is also visible for U and V, but not for W. But I agree that your modeling of the ABL is quite very similar to realistic conditions (the authors rigorously verify their set-up).

10. P12, L218-L19 : why do you show the spectral content for this case in particular ?

11. P12, L227-229 (discussion of Figure 9) : why do phi_max of points located at y/D = +-1 increases (compared to their nearest neighbour) and why do the authors conclude that this delimitates the wake bounds?

12. P13, L236-237 (discussion of Figure 10): "*This result is consistent with literature that finds a maximum of receptivity in the reduced frequency range of [0.2-0.3] (Li et al., 2021)*". The results of Figure 10 show that the peak is higher for fred > 0.3, so the maximum is not yet reached for 0.2 < fred < 0.3. This is not consistent with Li et al, 2021. Do the authors expect lower phi_max for higher fred ? or higher phi_max? The authors should be more critical about their results.

13. P16, Figure 12 : this figure is very unclear (same symbol and same color for different amplitudes, …).

14. P14-P15, discussion about realistic motion (1dof and 3dofs) : The authors could refer to the literature (i.e Belvasi et al, 2022) to enrich the discussion. Indeed, the authors mentioned in the introduction that Belvasi et al 2022 showed the impacts of the FOWT motion on wake statistics (centerline, velocity deficits, etc). Could the authors relate their results to those studied in the reference?

**Technical comments**

P1, L21-22 : A word is missing in the sentence? "*The authors state that both constructive and destructive XX is possible between the dominant scales in the wake and a potential downstream turbine,…*"

P3, figure : [H] on the right of the figure

P7, L150 : the authors should add the two notations (epsilon_uI and epsilon_sigmaU) in the text before the equations, and remove the point before the equations

P8, L182 : the notation z0 is not introduced (normally, the notation for the roughness length is well known, but this should be defined)

P12, L232 : the authors write *"For instance, the idealised characteristic 1-Dof motion regimes (cases S1, H1 and P1) … ".* The other cases presented in Figure 10 are also idealized. This sentence should be reformulated ("For instance, the idealized cases S1, H1 and P1 …")

Figures 10, 11 and 12 : phi_max and not phi in the labels

---

## Author Comment (AC1)

**REFEREE #1**

**General comments:**

"Floating wind turbine motions signature in the far-wake spectral content – A wind tunnel experiment" addresses the important question of the relationship between floating offshore wind turbine (FOWT) motions and wake meandering. Using an actuator disc model subject to idealized and realistic motions in a wind tunnel, the spectral signature of the wake is probed and related to the frequencies of the imposed motions. The authors find that the signature of strong, idealized motions appear in the wake, while more realistic motions do not have an apparent impact. While these findings present a step towards understanding wake meandering for FOWTs, the authors could have elaborated on their results to show what impacts realistic wave motions do have on the wake, even if the expected spectral signature was not observed. In addition, some of the details of both the methods and results were not clearly presented. Please see below for specific comments and some minor technical corrections.

The authors want to deeply thanks reviewer#1 for the careful review. The present document gives answers to each of the points raised. Reviewer remarks are in black, authors answers in blue. A "diff" file is also provided to enlighten the modifications made to the manuscript.

**Specific comments:**

1. Page 2, lines 44-46: Can you elaborate on how your work compares with Belvasi et al. (2022)? For example, page 16, line 288 states that none of the pitch regimes tested in the current study leave a detectable trace at 8D, but figure 13 of Belvasi et al. (2022) shows a clear peak for the case that is similar to P5 in the current study.

More description is made in the introduction to better understand the differences between the present work and Belvasi 2022.
In the result section, reference is made to cases SI and PII described in Belvasi 2022 as they are close to our cases S1 and P5.

Text added:
"At 8/D the peak magnitudes are roughly halved for surge motion. The peak for S1 is now visible, this is in agreement with Belvasi et al. (2022) where a peak is visible in the PSD of porous disc wake power for case SI where (0.125D/0.11), very similar to our case S1. It is to be noted that in the present study, we analyse the pre-multiplied PSD of the longitudinal wind speed component whereas in Belvasi et al. (2022) the PSD of the porous disc wake power is shown, the latter comprises spatial information of the wake and linked to the velocity at power 3, a direct comparison is therefore not straightforward."
and later:
"For pitch motion, the tested motion regimes do not deliver a visible peak in the normalised energy spectrum. In case P5, that is similar to the case PII in Belvasi et al. (2022) where a peak in the power is detected, the peak is too weak to be detected."
and in the conclusion:
"None of the motion regimes tested for heave and pitch leave a detectable trace at 8/D. However, Belvasi et al. (2022) showed peaks in the PSD of the wake power for a case similar to P5 suggesting that are if no clear peak is visible in the vertical centerline, the wake is still affected and more analysis using other approaches is needed."

2. Page 3, lines 61-62: Camp and Cal (2016) do not show wake spectra in their study. Is there evidence showing that the rotational frequency signature does not persist into the far wake? Aubrun et al. (2013), showed through wind tunnel experiments that no signature of the tip vortex remains visible at x/D = 3 for a neutral incoming boundary layer with a turbulence intensity of 13%. This conclusion was valid even for low turbulent inflow conditions (turbulence intensity 4%). These results are in agreement with Zhang et al. (2012, 2013) works. They showed that the tip vortex signatures are not distinguishable anymore from x/D >3 in a neutral or convective incoming boundary layer with a turbulence intensity of 8% at hub height. The text is modified to: "Several studies showed that tip vortex signatures at the edges of the wake are undetectable at x/D > 3 (Zhang et al. 2012, Zhang et al. 2013, Aubrun et al. 2013)."

Zhang, W., Markfort, C.D., Porté-Agel, F., 2012. Near-wake flow structure downwind of a wind turbine in a turbulent boundary layer. Experiments in Fluids 52, 1219–1235.

Zhang, W., Markfort, C.D., Porté-Agel, F., 2013. Wind turbine wakes in a convective boundary layer: wind tunnel study. Boundary Layer Meteorology 146, 161–179.

3. Page 4, lines 100-101: What does the velocity scaling factor of 3.2 correspond to in terms of the full-scale and model velocities used?
The full scale wind speed at hub height chosen is 8 m/s corresponding to the order of magnitude of the mean wind speed below 100m at the SEM-REV sea test site (see line 124-125).

4. Page 5, lines 121-129: The use of "first order" and "second order" is not clear in this context.
The text in section 2.3 (lines 114-130) is modified to include the definition of "first" and "second" order motion. Figure 7 was moved to this section to help the description. Modifications in the text:
"In the power spectral density of the surge, heave and pitch motion represented in Fig. 2 (OrcaFlex results), two peaks can be identified. The first peak near $f_{red}$ = 1 relates to the floater motion induced by wave-to-wave frequencies, it is called here first-order motion. The second peak at lower frequency, near $f_{red}$ = 0.1 is related to the response of the floater linked to the mooring lines and anchoring characteristics, it is called second-order motion."

5. Page 8, figure 3: Why don't the normalized velocity profiles reach 1 at the top of the plot?
In Figure 3, the free-stream velocity ($U_{ref}$) used for the normalization is well above the wind tunnel boundary layer, close to the middle of the test section. This is well outside of the figure, explaining why the $U/U_{ref}$ does not reach 1.
Added to the text line 186: "The normalised mean velocity profiles of each flow component are shown in Fig. 4. Notice that the free-stream velocity ($U_{ref}$) used for the normalization is well above the wind tunnel boundary layer, near the middle of the test section. For this reason $U/U_{ref}$ does not reach 1"

6. Page 8, line 175: Does the proposed range of 0.08 to 0.12 come from the VDI Guideline? Please clarify.
Yes it does, reference added.

7. Page 8, lines 176-177: Why is the flow slightly skewed to the left?
Added in the text line 194 "Probably due to inhomogeneities in the wind tunnel test section."

8. Page 8, line 181-182: The theoretical data is for z 0 = 0.01 m, which is still multiple orders of magnitude larger than the roughness length calculated in this experiment (z 0 = 5.5 × 10 −6 m). How is this comparison justified?
A new log fitting of Z_0 and U* was made leading to a new value of z_0 = 0.0057m (full scale) that is more consistent and well in the VDI range. Text is modified accordingly. The reference is removed (not relevant).

9. Page 10, figure 6: There are no blue dots (shown in the legend) plotted in the rightmost panel of this figure.
In fig 6, the legend refers to the turbulent spectra performed in the three directions. The spectra made in the wind tunnel are represented by blue dots and Kaimal reference by a continuous line and its associated reference range (color).
We double-checked the figure. All points and reference ranges should now be displayed correctly.

10. Pages 10-11, lines 209-210: This sentence is not very convincing as figure 7 shows only one point from Orcaflex below (or at) the cutoff frequency of 0.1 for each plot.
Indeed, the sentence is then modified to "... at the cut-off frequency…"

11. Page 12, lines 227-229: Why do the OL and OR points have more energy for the moving turbine than the fixed turbine (i.e., φ max > 0), and why does this signify the lateral limits of the wake?
Modifications in the text:
"Further, the outer points at OL and OR have slightly increased $\varphi_{max}$ compared to their respective neighbours. At these positions, the peak emerges more distinguishably from the local turbulence. This proves that these positions are located at the outer edge of the wake, where the turbulence variance decreases to tend to the external ambient turbulence, whereas the motion imposed to the overall wake is still energetic."

12. Page 13, lines 235-237: It does not appear that the results of this study are consistent with those from Li et al. (2021) which show maximum receptivity for reduced frequencies between 0.2 and 0.3. In the current study, the peak frequencies are all above 0.3. Were any higher frequencies tested? Would φ max be expected to continue to increase with frequency?
Li et al 2022 studied side-to-side motion (sway) using LES with an idealized uniform flow with low turbulence, a configuration rather far from our set-up. However very few results are available in the literature, so the comparison is kept. The original text was indeed badly written. A modification is proposed:

"In the limited number of configurations tested, we see a higher signature in the highest range tested close to $f_{red}$ = 0.3. Very few results are available in the literature to compare but, in their study of sway motion with idealized inflow conditions, Li et al 2022 showed a maximum of receptivity also in the range [0.2-0.3], but the local maxima they found is not visible in the present case."

Conclusions are also modified accordingly

13. Page 15, line 270: Does the realistic motion affect the total energy of the wake?
In the work of Belvasi 2022, the realistic motion does not significantly affect the TKE in the wake. This is added to the text.

14. Page 16, figure 12: This figure is very unclear. Are three cases shown on each plot? If so, different symbols/colors should be used to differentiate them.
The figure has been adapted for clarity. The different cases are now color-coded, with the shapes representing the measurement locations.

15. Page 17, lines 305-306: Does the realistic motion have any impact on the wake? Can you compare, for example, a time series of the wake centerline? Or could you refer to previous studies to put this finding in context?
In this work, we do not detect the wake centerline. Instead, we examine the spectral signature of the motion in the wake. Reference is made with Belvasi 2022 to put into perspective the complementarity of the results.

**Technical corrections:**
1. Page 1, line 22: A word is missing here: "...both constructive and destructive interference is possible..."
Indeed a word was missing, text modified to "The authors state that both constructive and destructive interference is possible between the dominant scales in the wake and a potential downstream turbine, highlighting the necessity to take this potentially dangerous interaction into account when designing FOWTs."

2. Page 5, line 123: If figure 7 is the first figure referenced, maybe it should be figure 1.
The figure is moved to section 2.3 and is better explained

3. Page 7, line 168-169: Make sure you are consistent with your use of $\varphi$ max vs. $\varphi$ max .
Changed to be consistent in the text and the figures.

4. Figures 9, 10, 11, and 12: The labels on these figures should read $\varphi$ max rather than $\varphi$.
The figures have been updated with the correct labels.

**REFEREE #2**

**General comment**
"Floating wind turbine motions signature in the far-wake spectral content – A wind tunnel experiment" investigates the relationship between a floating offshore wind turbine (FOWT) motion and wake dynamics. The authors consider a realistic FOWT motion, and they want to compare its impact on the wake signature to that obtained using idealized motions (harmonics, etc).

For this research work, they perform wind tunnel experiments with a porous Actuator Disk. The model is equipped with a motor that allows to represent heave, surge and pitch motions of the floater (the other motions are negligible for the considered "wave" case). When idealized motions are considered, the signature of those motions is visible in the spectral content of the wake velocity fluctuations (when high amplitudes and high frequencies are imposed). When a realistic FOWT motion is considered, the impact of this motion on the wake power spectra is negligible. The authors conclude that one should be careful by using idealized sinusoidal signals for the FOWT motions as their impacts on the wake signature are different from those obtained using a more realistic signal (at least for the considered size of wind turbine, i.e. D = 80 m).
The paper is well written and pleasant to read. The research work is very interesting, as it clearly highlights differences in terms of wake response between realistic and idealized conditions. The objectives are clearly highlighted, and the study is very rigorous.
However, they authors could more elaborate on the impact of the realistic motion on the wake dynamics in general. Some paragraphs could also be clearer, and some results should more clearly presented or more discussed. Please see below for more specific comments

The authors want to deeply thanks reviewer#2 for the careful review. The present document gives answers to each of the points raised. Reviewer remarks are in black, authors answer in blue. A "diff" file is also provided to enlighten the modifications made to the text.

**Specific comments**
1. P2,L30 --- : I am surprise that there are no more LES studies that focus on the impact of the wind turbine motions on the wake signature. Or perhaps, if they do exist, these studies always consider idealized motions?
Indeed the vast majority of the literature using LES have idealized inlet conditions and idealized motion, but the number of studies analyzing numerically the far wake effect of turbine motion is rather limited. However, we have added the study of Kleine 2022, which is relevant for our study.

2. P2, L45 – L50 : comparison with work of Belvasi et al. (2022) The authors have exactly the same set-up? The work of Belvasi et al. (2022) also proposes to investigate differences in wake statistics between idealized and realistic motions? What does this present study add? According to the authors, the work of Belvasi et al. (2022) already highlighted that realistic FOWT motions have an impact on the wake center position, and not on the wake deficit, and on the TKE. The authors should highlight more clearly what this "complementary" research paper will add.

The second part of the introduction was modified to better understand the similarities/differences compared to Belvasi 2022. The experimental set-up is the same but the instrumentation differs. In Belvasi 2022, only harmonic motions were studied on a limited number of cases (5) compared to 19 in the present paper.
The results of Belvasi 2022 were compared at several places in the text.

3. P3, L67 -> L70 : In this paragraph, both motions should be mentioned (idealized and realistic). The authors also write that they will only investigate heave, surge and pitch, and it is not clear at this stage why they consider these only.
It is clarified in the introduction section that both idealized and realistic motion are tested and that surge, pitch and heave are chosen because they represent platform motions in aligned wind/wave conditions.

4. P4-P5, section 2.2 : The authors should give an order of magnitude of the wind speed considered in this research work (both for the wind tunnel and the FOWT).
The full scale wind speed at hub height chosen is 8 m/s corresponding to the order of magnitude of the mean wind speed below 100m at the SEM-REV sea test site (see line 124-125 in the original version, and line 129-131 in the revised version with modification tracking).

5. P4-P5 : An image of the experimental set-up (porous disk, ...) should be useful (even if it is surely available in other papers).
An image of the experimental set-up is included showing the porous disk.

6. P5, Section motion modeling (2.3) : It is not clear for me : the authors have motion signals coming from Orcaflex, for a particular wave case, and they will use this signal as a realistic motion ("realistic case" of the paper), right? And this "proposed range" come from?
The proposed range comes from the VDI guideline. The relevant reference is added.

9. P8, L187-188 (discussion of Figure 6) : "Nevertheless, even if the experimental spectra are within the range of the reference spectra provided in VDI (2000) (shaded zones), t, based on this signal, the authors will also isolate characteristic amplitude and frequencies in order to elaborate idealized floater motions? This last part should be more elaborated or described in Section 2.3. This becomes clear later in the paper but this is not the case in this paragraph.
We used a reference signal coming from an Orcaflex simulation performed by the manufacturer of the floater (which is satisfyingly matching full-scale motion, Choisnet et al 2018) as a reference for the "realistic" motion. The simulation was performed for given wind/wave conditions representative of the site. The reference signal was also used to identify harmonic amplitudes and frequencies to be reproduced in the wind tunnel by the motion system. The paragraph is slightly modified and the figure presenting spectra of the motion (fig 7 in original version), is moved to this section for better understanding.

7. P5, L121: What do "first order" and "second order" mean?
The text in section 2.3 (lines 114-130) is modified to include the definition of "first" and "second" order motion. Figure 7 was moved to this section to help the description. Modification in the text:
"In the power spectral density of the surge, heave and pitch motion represented in Fig. 2 (OrcaFlex results), two peaks can be identified. The first peak near $f_{red}$ = 1 relates to the floater motion induced by wave-to-wave frequencies, it is called here first order motion. The second

peak at lower frequency, near $f_{red}$ = 0.1, is related to the response of the floater linked to the mooring lines and anchoring characteristics, it is called second order motion."

8. P8, L175 : "The corresponding power-law profile exponent α is 0.11, which is within the proposed range between 0.08 and 0.12." Where doeshey are shifted towards higher frequencies compared to the Kaimal model". What do the authors conclude about this shift? As the sentence is formulated, the reader expects a conclusion. This shift is also visible for U and V, but not for W. But I agree that your modeling of the ABL is quite very similar to realistic conditions (the authors rigorously verify their set-up).

No real consequences are expected from this shift. The text is modified to:

"Nevertheless, even if the experimental spectra are within the range of the reference spectra provided in VDI (2000) (shaded zones), they are slightly shifted towards higher frequencies compared to the Kaimal model. The shift  implies that, at wind-tunnel scale, the peak of energy happens for slightly smaller turbulent structures than expected in real life, but deviations are too limited to expect an impact on the physics described in this paper.."

10. P12, L218-L19 : why do you show the spectral content for this case in particular ?

We think it is of interest to show, at least once, the PDF in the wake of the fixed and moving model. After this graph, only $\varphi_{max}$ are presented.

Modification in the text:

"Figure 8 shows an example of the pre-multipli…."

11. P12, L227-229 (discussion of Figure 9) : why do phi_max of points located at y/D = +-1 increases (compared to their nearest neighbour) and why do the authors conclude that this delimitates the wake bounds?

Modifications in the text:

"Further, the outer points at OL and OR have slightly increased $\varphi_{max}$ compared to their respective neighbours. At these positions, the peak emerges more distinguishably from the local turbulence. This proves that these positions are located at the outer edge of the wake, where the turbulence variance decreases to tend to the external ambient turbulence, whereas the motion imposed to the overall wake is still energetic."

12. P13, L236-237 (discussion of Figure 10): "This result is consistent with literature that finds a maximum of receptivity in the reduced frequency range of [0.2-0.3] (Li et al., 2021)". The results of Figure 10 show that the peak is higher for fred > 0.3, so the maximum is not yet reached for 0.2 < fred < 0.3. This is not consistent with Li et al, 2021. Do the authors expect lower phi_max for higher fred ? or higher phi_max? The authors should be more critical about their results.

Li et al 2022 studied side-to-side motion (sway) using LES with an idealized uniform flow with low turbulence, a configuration rather far from our set-up. However very few results are available in the literature, so the comparison is kept. The original text was indeed badly written. A modification is proposed:

"In the limited number of configurations tested, we see a higher signature in the highest range tested close to $f_{red}$ = 0.3. Very few results are available in the literature to compare but, in their study of  sway motion with idealized inflow conditions, Li et al. (2022) showed a maximum of

receptivity also in the range [0.2-0.3], but the local maxima they found is not visible in the present case."

Conclusions are also modified accordingly

13. P16, Figure 12 : this figure is very unclear (same symbol and same color for different amplitudes, ...).
The figure has been adapted for clarity. The different cases are now color-coded, with the shapes representing the measurement locations.

14. P14-P15, discussion about realistic motion (1dof and 3dofs) : The authors could refer to the literature (i.e Belvasi et al, 2022) to enrich the discussion. Indeed, the authors mentioned in the introduction that Belvasi et al 2022 showed the impacts of the FOWT motion on wake statistics (centerline, velocity deficits, etc). Could the authors relate their results to those studied in the reference?
Reference is made with Belvasi 2022 to put into perspective the complementarity of the results.
In general, more comparison to Belvasi 2022 is performed.

**Technical comments**
P1, L21-22 : A word is missing in the sentence? "The authors state that both constructive and destructive XX is possible between the dominant scales in the wake and a potential downstream turbine,..."
Indeed a word was missing, text modified to "The authors state that both constructive and destructive interference is possible between the dominant scales in the wake and a potential downstream turbine, highlighting the necessity to take this potentially dangerous interaction into account when designing FOWTs."

P3, figure : [H] on the right of the figure
Removed.

P7, L150 : the authors should add the two notations (epsilon_uI and epsilon_sigmaU) in the text before the equations, and remove the point before the equations
Notations added to the text.

P8, L182 : the notation z0 is not introduced (normally, the notation for the roughness length is well known, but this should be defined)
Notation introduced in section 2.5.

P12, L232 : the authors write "For instance, the idealised characteristic 1-Dof motion regimes (cases S1, H1 and P1) ... ". The other cases presented in Figure 10 are also idealized. This sentence should be reformulated ("For instance, the idealized cases S1, H1 and P1 ...")
The text has been adapted accordingly.

Figures 10, 11 and 12 : phi_max and not phi in the labels

The figures have been updated with the correct labels.

---

## Referee Report (RR1)

**General comments:**

The authors' changes have substantially improved the clarity of the manuscript. All that remains are a few technical corrections, mostly in the modified text.

**Technical corrections:**

1. Page 11, lines 190-192: $z_0 = 5.7 \times 10^{-3}$m is not within the range of $1 \times 10^{-5}$m $\leq z_0 \leq 5 \times 10^{-3}$m. Did the authors intend to indicate that $z_0$ was close to the guideline range, or were they referring to the wind-tunnel scale value of $z_0 = 1.15 \times 10^{-5}$m?
2. Pages 14-15, lines 270-273: Please split this sentence into two sentences to improve readability. A couple words also appear to be missing: "…is linked to the velocity at power 3, *so* a direct *comparison* is not straightforward."
3. Page 20, line 321: There is an extra word in this sentence: "…suggesting that  if no clear peak is visible…"

---

## Author Response (AR2)

wes-2023-105-referee-report-2

General comments:
The authors' changes have substantially improved the clarity of the manuscript. All that remains are a few
technical corrections, mostly in the modified text.

Technical corrections:
1. Page 11, lines 190-192: $z_0 = 5.7 \times 10^{-3}$ m is not within the range of $1 \times 10^{-5}$ m $\leq z_0 \leq 5 \times 10^{-3}$ m. Did the authors intend to indicate that $z_0$ was close to the guideline range, or were they referring to the wind-tunnel scale value of $z_0 = 1.15 \times 10^{-5}$ m?

Z0 is very close to the VDI range. This is corrected in the revised version:
"At full scale, it gives $z_0 = 5.7 \times 10^{-3}$m that is very close to the VDI Guideline's range [$1 \times 10^{-5}$ m - $5 \times 10^{-3}$ m]."

2. Pages 14-15, lines 270-273: Please split this sentence into two sentences to improve readability. A couple words also appear to be missing: "...is linked to the velocity at power 3, so a direct comparison is not straightforward."

This part is modified to:
"At that distance, no peaks are detected for heave and pitch motions anymore (Fig. \ref{fig:rel_max_8}). In the present study, the pre-multiplied PSD of the longitudinal wind speed component is analyzed. In contrast, in \citep{Belvasi2022} the PSD of the porous disc wake power is shown, which comprises spatial information of the wake and is linked to the velocity at power 3. Therefore, a direct comparison with the present work is not straightforward."

3. Page 20, line 321: There is an extra word in this sentence: "...suggesting that are if no clear peak is visible..."

modification done